# Formation and Investigation of Electrospun Eudragit E100/Oregano Mats

**DOI:** 10.3390/molecules24030628

**Published:** 2019-02-11

**Authors:** Juste Baranauskaite, Erika Adomavičiūtė, Virginija Jankauskaitė, Mindaugas Marksa, Zita Barsteigienė, Jurga Bernatoniene

**Affiliations:** 1Institute of Pharmaceutical Technologies, Lithuanian University of Health Sciences, Medical Academy, Sukileliu pr. 13, LT-50162 Kaunas, Lithuania; jurga.bernatoniene@lsmuni.lt; 2Department of Analytical and Toxicological Chemistry, Lithuanian University of Health Sciences, Medical Academy, A. Mickeviciaus g. 9, LT-44307 Kaunas, Lithuania; mindaugas.m.lsmu@gmail.com; 3Department of Production Engineering, Faculty of Mechanical Engineering and Design, Kaunas University of Technology, Studentu st. 56, LT-51424 Kaunas, Lithuania; erika.adomaviciute@ktu.lt (E.A.); virginija.jankauskaite@ktu.lt (V.J.); 4Department of Pharmacognosy, Lithuanian University of Health Sciences, Medical Academy, A. Mickeviciaus g. 9, LT-44307 Kaunas, Lithuania; zita.barsteigiene@lsmuni.lt; 5Department of Drugs Technology and Social Pharmacy, Lithuanian University of Health Sciences, Medical Academy, A. Mickeviciaus g. 9, LT-44307 Kaunas, Lithuania

**Keywords:** electrospun, rosmarinic acid, carvacrol

## Abstract

An electrospun mat of Eudragit E100 (EE100) (a cationic copolymer based on dimethylaminoethyl methacrylate, butyl methacrylate, and methyl methacrylate) was used as a delivery system for oregano ethanolic extract (OEE). Oregano is a biologically active material which is widely used because of the antibacterial and antifungal activity. The oregano herb consists of phenolic compounds, the main of which are rosmarinic acid and from essential oil—carvacrol. Such a material could be an ideal candidate for oral drug systems. The influence of the EE100 concentration in the OEE on the structure of electrospun mats, encapsulation efficiency, dissolution profile, release kinetics and the stability of biologically active compounds was investigated. The concentration of the solution is a critical parameter for the structure and properties of electrospun mats. The diameter of electrospun fibers increased with the increase of EE100 concentration in the OEE. Electrospun mats obtained from 24% to 32% EE100 solutions showed high encapsulation efficiency, quick release and high stability of rosmarinic acid and carvacrol. Dissolution tests showed that 99% of carvacrol and 80% of rosmarinic acid were released after 10 min from electrospun nano-microfiber mats and capsules obtained from such formulations. The stability tests showed that physicochemical properties, dissolution profiles, and rosmarinic acid and carvacrol contents of the formulations were not significantly affected by storage.

## 1. Introduction

Many recent investigations have revealed benefits of natural products for human health. Natural compounds from plants demonstrate various biological effects, including antibacterial, antimalarial, antifungal and antioxidant activity [1]. Their biological activity is attributed to the presence of phenolic compounds and essential oils, which are the main constituents of multiple pharmacologically active products [2]. Turkish oregano (*Origanum onites* L.) (hereinafter referred to as oregano) is rich in phenolic compounds, the main of them being rosmarinic acid, with powerful antibacterial and antifungal properties [3,4]. Furthermore, oregano essential (OE) oils have been shown to possess antioxidant, antibacterial, antifungal, diaphoretic, carminative, antispasmodic, and analgesic activities [5]. In recent years, the efficiency of essential oils from oregano species has been reported in many studies. Basser et al. identified carvacrol as the main phenolic compound responsible for biological activities of oregano [6]. However, various natural products have limited bioavailability due to the low solubility, which affects their absorption by human body. Furthermore, stability issues of natural products like hydrolysis, oxidation, and photolysis urge the need for stabilization. The limited solubility and stability problems can be controlled by polymer nanoparticles using different carrier categories [7,8].

With the increasing popularity of nanotechnology, electrospun nano-microfibrous materials have drawn attention to their potential applications in different biomedical areas, such as drug delivery, wound dressing, scaffolds, minitablets, tablets, and cosmetic masks [9,10,11]. Electrospun nano-microfibers are used as drug carriers in the drug delivery systems due to their highly functional characteristics and different controlled drug release profiles such as immediate, sustained, and biphasic release. The preparation of electrospun nano-microfiber materials has attracted particular interest because of its simplicity and low costs, the porosity and high surface area-to-volume ratios of the fibers, and the wide variety of materials, which can be processed [12,13]. Moreover, the fibers have high encapsulation efficiency, as there is no loss of active substance during the preparation [14,15,16,17]. 

Synthetic polymers such as polycaprolactone, polyvinylpyrrolidone and Eudragit S100, commonly used in electrospinning, provide great flexibility in synthesis, processing and modification of nano-microfibers materials and possess good mechanical properties and stability in the body [18].

For the preparation of electrospun nano-microfiber mats as a polymer carrier, Eudragit E100 (EE100) has been selected. EE100 is a pH-dependent polymer, soluble in gastric fluid and swelling at pH lower than 5.0. It has low viscosity and high pigment biding and can be applied in different pharmaceutical dosage forms. EE100 can be used as a film/insulation coating material, which is suitable for taste/odor masking and light/moisture protection [19]. Moreover, it can be used in nanoparticles production by blending polylactic glycolic acid and EE100 for plasmid delivery [20], [21].

Various polymeric nano-microfibers have been already produced by electrospinning techniques. Therefore, active agent encapsulating polymeric films and electrospun nano-microfibers have been studied previously. Munhuweyi with co-authors [22] have investigated physical and antimicrobial activities of encapsulated cinnamon and OE oils using β-cyclodextrin (β-CD) and electrospun chitosan/polyvinylalcohol/β-CD/OE nanofibrous films [22].

Moreover, the electrospinning technique has been described in several articles as the potential method of formation of drug-loaded electrospun nano-microfibers, used as a new solid matrix for solid dosage forms. Several advantages have been identified of electrospun nano-microfibers used for drug delivery systems: high porosity of these structures can facilitate fast wetting, which promotes a rapid release of the drug. Moreover, drugs can be incorporated into electrospun nano-microfibers in an amorphous form, which can be utilized to increase both the apparent solubility and the dissolution rate of the drug [13,23]. Hamori with co-authors [24] have prepared nano-microfiber-based capsules for oral use including uranine and nifedipine drugs for a controlled release delivery system using methacrylic acid copolymer (MAC, Eudragit^®^ S100). The in vitro release tests of uranine and nifedipine from the nanofiber packed capsules and milled powder of nanofiber packed capsules showed efficient results compared to capsules of a physical mixture of MAC and each drug. This study investigates the potential of natural active compounds-loaded electrospun nano-microfiber mats as a new solid matrix for capsules. The influence of EE100 concentration in oregano extract on the structure, encapsulation efficiency, stability, dissolution profile and release of biologically active compounds from electrospun mats was investigated. 

## 2. Results and Disscusion

### 2.1. Structure Analysis

It is well known [25] that the structure of electrospun mats depends upon polymer and solvent nature, solution properties (viscosity, conductivity, and surface tension), processing parameters (applied voltage, and distance between electrodes) and electrospinning environmental parameters. In this study, the content of EE100 in the oregano ethanolic extract (OEE) was varied and sample compositions are shown in Table 1. From the data presented in Table 1 and SEM images in Figure 1, it can be seen that the structure of electrospun mats and the diameter of electrospun fibers depended on the content of EE 100 in OEE in an electrospinning solution. In the case of E-8 sample, the EE100 polymer content was too small to obtain electrospun fibers of required quality. Therefore, in this case, after the eletrospraying process, only polymer spots were formed on the support material (Figure 1A). Increased electrospinning solution viscosity, and decreased conductivity with increase of the OEE content influenced the increase of electrospun fibers diameter (Table 1 and Figure 1A–E). With the content of 40 wt. % of EE100 in the electrospinning solution (E-40 sample), the average diameter of electrospun fibers increased 6 times, compared to that of E-16 sample.

Analysis of SEM images showed that electrospun mats obtained from the solutions of OEE with 16 wt. % and 24 wt. % of EE100 (E-16 and E-24) consisted of uniform, thin nano-microfibers (average diameters of 200 ± 67 nm and 326 ± 110 nm, respectively). The mats, formed from microfibers with average diameters of 1210 ± 508 nm and 3101 ± 1120 nm, respectively, were obtained from electrospinning solutions with higher EE100 content (Figure 1D,E).

It was determined that 99% and 76% of the fibers had a diameter up to 400 nm when contents of EE100 in OEE electrospinning solutions were 16 wt. % and 24 wt. %, respectively (E-16, and E-24, Figure 2A). The thinnest fibers have been formed from the 16 wt. % EE100 solution (E-16). It was determined that 61% and 7% of nano-microfibers had a diameter up to 200 nm, when EE100 solutions with respective contents of EE100 of 16 wt. % and 24 wt. % were used (E-16, and E-24, Figure 2B).

No chemical interaction was observed between EE100/OEE mat components. Only hydrogen interactions between OH groups and also between OH and NH groups of EE100 and OEE active compounds were detected (not shown). The formation of intermolecular hydrogen bonds in the presence of NH groups is more favorable than that in the presence of NH_2_ groups.

### 2.2. Encapsulation Efficiency (EE) Determination

EE is an important parameter of formulation. The amount of active compounds (load) in a given weight of nano-microfibers can be found from the formulation’s composition, when the extract is fully encapsulated by the polymer. However, in practice, the materials used to prepare nano-microfibers formulation would not be fully encapsulated. The actual amounts of rosmarinic acid and carvacrol in prepared nano-microfiber mats were determined by high-performance liquid chromatography. As illustrated in Table 2, the E-32 formulation of electrospun mats (the diameter of fibers varied from 400 to 2400 nm (Figure 2A)) was the best in terms of EE of rosmarinic acid (85.6%) and carvacrol (91.8%). Furthermore, the electrospun mats of E-24 formulation also showed high EE values for rosmarinic acid (70.45%) and carvacrol (87.7%) and the diameter of fibers was nano-microsized with the average fiber diameter of 326 ± 110 nm, whereas the E-16 formulation electrospun mats (the diameter of fibers varied from 150 to 400 nm, the average fiber diameter: 200 ± 68 nm) had the lowest EE of rosmarinic acid (50.23%) and carvacrol (61.91%). It could be related to the content of EE100 in the OEE solution being too low fully cover the active compounds and resulting in a decreased EE. The relatively high EE of carvacrol is related to the hydrophobic nature of polymer and carvacrol. However, the successful encapsulation of the water-soluble agents (rosmarinic acid) in the hydrophobic polymers (EE100) still presents challenges in high drug loading and their denaturation during the formulation process [26,27]. As compared to our previous studies, where oregano extract microcapsules were prepared by a spray-drying technique and the wall materials were gumi arabic and maltodextrin, the EE has increased 12.9 times for carvacrol and 2.4 times for rosmarinic acid (*p* < 0.05) using a polymer electrospining method [28]. The result shows that polymers make strong core–shell structure to increase the EE.

### 2.3. In Vitro Drug Release

According to EE results, in vitro drug release tests were used for E-24 and E-32 formulations. The quality of produced nano-microfibers mats and prepared capsules by using milling was defined by the release profile of OEE active compounds rosmarinic acid and carvacrol. The dissolution profiles of nano-microfibers mats and capsules of formulations E-24 and E-32 are shown in Figure 3. The concentrations were determined by extrapolation of the calibration curve, and a graph of present release over time was plotted.

The results showed that electrospun nano-microfiber mats had 2.5 times higher dissolution profile of carvacrol in the first 3 min compared to that of the capsules (Figure 3). Moreover, it was observed from the dissolution data that more than 75% of the carvacrol was released from the E-24 and E-32 formulations nano-microfiber mats and capsules during the first 5 min. The in vitro data of carvacrol release from E-24 and E-32 formulations mats and capsules showed that about 100% of the active compounds were released after 10 min. It was revealed that there were no significant differences between E-24 and E-32 formulations fiber mats and capsules (Figure 3). High dissolution data influenced the choice of EE100 as a drug carrier, which plays an important role in drug delivery systems. EE100 has low viscosity, high pigment binding capacity, good adhesion and low polymer weight gain, which enhance the bioavailability of active compounds (carvacrol is a monoterpenoid phenol), in particular those with low aqueous solubility [29]. Moreover, the fast dissolution rate of electrospun nano-microfiber mats has been attributed to a number of properties, which include the hydrophobic nature of polymer and carvacrol, the high surface area of the nano-microfiber matrix structure and the molecular dispersion of the carvacrol [30].

In Figure 3, the results of dissolution tests performed to compare release of rosmarinic acid from the E-24 and E-32 formulations nano-microfiber mats and capsules are presented. The same tendency was observed with rosmarinic acid: electrospun nano-microfibers mats had significantly 5.0 times higher dissolution profile of rosmarinic acid after first 3 min as compared to formulations capsules (Figure 3). The slower rosmarinic acid release from the capsules is presumably due to the lower specific surface area exposed to the media compared to the electrospun mat. In capsules, the formulation may affect the network porosity influencing the capacity of the dissolution media to diffuse into the capsule matrix [30]. The in vitro data of rosmarinic acid release from electrospun nano-microfibers mats and capsules showed that about 78% of the active compounds were released after 10 min. Lower release results are affected by structure interactions between polymer and rosmarinic acid. Rosmarinic acid is the ester of caffeic acid and 3,4-dihydroxyphenyl lactic acid. EE100 contains strong acceptor groups (tertiary amines), but has less hydrogen-bonding potential. EE100 forms hydrogen bonds with rosmarinic acid through phenolic OH interactions with its dimethylamin groups, but because of the low number of OH groups in the polyphenol structure, it is complicated to form strong bonds [31].

Consequently, it was revealed that there was no significant difference between mat and capsule fillings after milling. The similar results were obtained in References [30,31].

### 2.4. Release Kinetics

The release versus time profiles from E-24 and E-32 samples electrospun nano-microfibers mats and capsules were shape-fitted against zero-order, first-order and Korsmeyer–Peppas kinetic models. The correlation was best reflected by Korsmeyer–Peppas and first-order models (Table 3). The established zero-order and first-order mathematical models, accordingly, were fitted to the drug release versus time profiles, in hope to better elucidate the kinetics of rosmarinic acid and carcvacrol release (Table 3). First-order results signified the independence of rate versus the concentrations of active compounds along each side of the membrane. In practice, such findings more often reflect a pseudo first-order model, in which dissolution rates are concentration-dependent. Regular time-release kinetics, for instance, would impart substantial advantages to drug delivery in medicine, and agriculture. Hence, the findings in Table 3 also imply that E-24 and E-32 could serve as reservoir systems for the continuous delivery of encapsulated carvacrol and rosmarinic acid.

The release mechanisms of carvacrol and rosmarinic acid from electrospun mats and capsules were analyzed using the Korsmeyer–Peppas model [31]. This model concerns the release of drugs from cylindrical structures and predicts whether the release of the compound from a matrix follows Fickian diffusion, through determination of the coefficient “n” estimated from the linear regression of the log (cumulative release) as a function of log (time). The “n” determined from both release curves was above 0.45, with a correlation value (R**^2^**) of 0.98 and 0.97 for carvacrol and rosmarinic acid, respectively. The release mechanism of both bioactives was mainly due to the swelling of the nano-microfibers that influenced their release. Wongsasulak and co-authors [31,32] also reported that the swelling of the matrix (electrospun polymers of zein, poly(ethylene oxide), and chitosan) triggered the release of the bioactive (α-tocopherol) according to the Korsmeyer–Peppas model.

### 2.5. Stability Studies

The stability studies of electrospun nano-microfiber mats (E-24 and E-32 samples) and capsules included the organoleptic evaluation, average weight of capsule contents, rosmarinic acid and carvacrol stability in formulations and their dissolution tests under long-term conditions for 6 months and accelerated stress conditions for 6 months. Table 4 demonstrates the stability test results of different formulations. The formulations did not show any physical changes during the study period.

Furthermore, the content of rosmarinic acid and carvacrol in the E-24 and E-32 formulations nano-microfibers mat and capsules after storage had no significant differences as compared with the fresh ones. The contents of rosmarinic acid and carvacrol in all formulations were more than 99 ± 0.5%.

The results revealed that there were no significant changes (*p* > 0.05) of dissolution rate of rosmarinic acid after aging capsules and microfiber mats under accelerated stress conditions (temperature: 40 ± 2 °C, relative humidity: 75 ± 5%) and long-term conditions (temperature: 25 ± 2 °C, relative humidity: 60 ± 5%).

The results mentioned above could be explained by interactions between active compounds and EE100 molecular structures. As it was mentioned above, EE100 ionic interactions play a significant role in both drug–polymer miscibility and active compounds physical stability, because of flavonoid polymer hydrogen bonding forces in the system [3,17,33].

Results are presented mean ± SD (*n* = 6, *p* ≥ 0.05). No significant differences between compared groups (E-24 nano-microfiber mats, E-32 nano-microfiber mats, E-24 nano-microfiber capsules, and E-32 nano-microfiber capsules) in different time periods (initial, 3 months, and 6 months) of carvacrol and rosmarinic acid content were observed.

## 3. Materials and Methods

Dried *Origanum onites* L. herb was obtained from “İnanTarım ECO DAB’’, Antalya, Turkey. The Herbarium of the Department of Drug Technology and Social Pharmacy, Lithuanian University of Health Sciences holds the voucher specimens (No. L170711). Ethanol (96%) for extraction was purchased from Vilniaus degtinė (Vilnius, Lithuania). EE100 was supplied from Evonik (Essen, North Rhine-Westphalia, Germany). The water used in HPLC and for sample preparation was produced with a Super Purity Water System (Millipore, Bedford, MA, USA). HPLC eluents: methanol (99.95%) was purchased from Carl Roth GmbH (Karlsruhe, Germany) and acetic acid (99.8%) from Sigma-Aldrich (St. Louis, MO, USA). Standards for HPLC analysis: carvacrol (>98%) was purchased from Sigma-Aldrich (St. Louis, MO, USA) and rosmarinic acid (>98%) was from ChromaDex (Santa Ana, TX, USA).

### 3.1. Preparation of Oregano Ethanol Extract

Prior to the extract preparation, oregano herb was ground in a cross beater mill IKA A11 Basic Grinder (IKA Works, Staufen, Germany) and sieved using a vibratory sieve shaker AS 200 basic (Retsch, UK) equipped with a 125µm sieve. The powdered material (100 g) was extracted with 1000 mL of 90% (*v*/*v*) ethanol in a round bottom flask by heat-reflux extraction performed in a water bath Memmert WNB7 (Memmert GmbH & Co. KG, Schwabach, Germany) at 95 °C for 4 h. The prepared extract was filtered using a vacuum filter. These conditions were determined as the best for the extraction of main active compounds of Turkish oregano in our previous study [34].

### 3.2. Preparation of Electrospinning Solutions

Various amounts of the EE100 granules were soaked in the OEE and magnetically stirred for 24 h at room temperature in order to obtain homogenously dissolved solutions. The polymer amounts in solutions were 8 wt. %, 16 wt. %, 24 wt. %, 32 wt. % and 40 wt. %. The solutions were degassed without stirring for 1 h prior to electrospinning.

### 3.3. Viscosity, and Conductivity of the Electrospinning Solution

The viscosity of EE100 solution in the OEE was measured by a rotational viscometer Alpha series (Fungilab, Barcelona, Spain) equipped with L1 spindles at 25 °C and a 10 rpm shear rate. The measurement was taken after 10 s. The conductivity of EE100 solution in the OEE was measured by the HQ40d portable multi meter (Hach, Loveland, CO, USA) at room temperature (25 ± 1 °C).

### 3.4. Electrospinning Process

The materials from nano-microfibers were formed using “Nanospider^TM^“ (Elmarco, Liberec, Czech Republic) electrospinning equipment. Electrospun mats from EE100/OEE solution formation parameters were following: applied voltage *U* = 70 kV, distance between electrodes *L* = 13 cm, temperature of environment *T* = 19 ± 2 °C and relative humidity *φ* = 40 ± 4%. Fiber-collecting time on the support material was 6 min.

### 3.5. Characterization of Electrospun Mats

The morphology of electrospun mats was determined using a scanning electron microscope SEM S-3400N (Hitachi, Tokyo, Japan). SEM images (magnification: 10,000×, scale bar: 5 µm; magnification: 1000×, scale bar: 50 µm; and magnification: 100×, scale bar: 500 µm) were analyzed. The diameter of nano-microfibers was evaluated using SEM images and software NIS-Elements D (Nikon Corporation, Tokyo, Japan). The average of nano-microfibers diameter was calculated from 100 measurements of SEM images (magnification: 10,000×, scale bar: 5 µm).

### 3.6. Quantitative Analyses

High-performance liquid chromatography (HPLC) analysis was carried out using Waters 2695 chromatography system (Waters, Milford, MA, USA) equipped with Waters 996 PDA detector. Data were collected and analyzed using a PC and the Empower 2 chromatographic manager system (Waters Corporation, Milford, USA). For determination of rosmarinic acid and carvacrol, an ACE 5 C18 250 × 4.6 mm column (Advanced Chromatography Technologies, Aberdeen, Scotland) was used.

#### 3.6.1. Preparation of Electrospun Mats for HPLC Analysis

For determination of rosmarinic acid and carvacrol, 50 mg of the prepared electrospun mats were accurately weighed and dispersed in 10 mL methanol in a volumetric flask and extracted for 10 min in an ultrasound bath (Memmert WNB7 water bath, Memmert GmbH & Co. KG, Schwabach, Germany). The prepared solution passed through a 0.45 µm diameter membrane filter for HPLC analysis.

#### 3.6.2. HPLC Conditions for Determination of Rosmarinic Acid

The mobile phase was composed of solvent A (methanol) and solvent B (0.5% (*v*/*v*) acetic acid in water). The following linear gradient elution profile was used: 95% A/5% B—0 min, 40% A/60% B—40 min, 10% A/90% B—41–55 min, 95% A/5% B—56 min. The flow rate was 1 mL/min and the injection volume was 10 μL. The effluent was determined at a wavelength of 329 nm. The quantification was carried out by the external standard method. The linear calibration curve was made (R^2^ = 0.999), and the peak areas were used for quantification [34].

#### 3.6.3. HPLC Conditions for Determination of Carvacrol

For determination of carvacrol, an ACE 5 C18 250 × 4.6 mm column (Advanced Chromatography Technologies, Aberdeen, Scotland) was used. The mobile phase was composed of methanol and water (60/40, *v*/*v*). The flow rate was 0.6 mL/min and the injection volume was 10 μL. The absorption was measured at 275 nm. The quantification was carried out by external standard method. The calibration curve was made (R^2^ = 0.999) [34].

### 3.7. Encapsulation Efficiency of Rosmarinic Acid and Carvacrol

EE, also known as active retention, is defined as the ratio of the concentration of encapsulated active ingredient (practical load) to its initial concentration (theoretical load) at the beginning of the encapsulation process. It was calculated by the equations adopted from Panda et al. [35,36]:(1)$$\mathrm{EE}\;\left(\%\right)=\frac{\mathrm{Practical}\;\mathrm{load}}{\mathrm{Theoretical}\;\mathrm{load}}\times \;100\;$$

### 3.8. Preparation of Capsule Fillings from Prepared Nano-Microfiber Mats

For preparation of powder, nano-microfiber mats were crushed by a PM-100 planetary mill (Retsch Co., Ltd., Germany) by adding 10 g of zirconium balls with a diameter of 1.0 mm by using a rotation speed of 300 rpm, and the procedure was performed for 3 h. The milled powder was then filled into number five gelatin capsules [32]. The capsules were filled using the manual capsule-filling machine (Capsuline, Pompano Beach, FL, USA).

### 3.9. Determination of Dissolution Profiles and Their Variability

Dissolution profiles of the prepared capsules and mats were determined with SOTAX AT 7 smart (Sotax, Allschwil, Switzerland) using the basket method at 50 rpm in 500 mL of artificial gastric juice (AGJ) without pepsin with a pH value of 1.5 at 37 ± 0.5 °C. Five milliliters of samples were taken from dissolution vessels manually at the time points of 1, 3, 5, 7, 10, 15, 20, 25, 30, 45 and 60 min, filtered through a 0.45 µm disposable filters and analyzed by HPLC for determination of active compounds. The dissolution media was then replaced by 5 mL of the fresh dissolution fluid to maintain a constant volume. The mean value of nanoparticles of six samples and a standard deviation for each sample were calculated. The evaluation of dissolution profiles was carried out in triplicate.

### 3.10. Stability Studies

Four ounce amber glass containers, each containing 30 capsules and microfibers mats, were kept according to The International Council for Harmonisation of Technical Requirements for Pharmaceuticals for Human Use (ICH) guidelines for the long-term storage conditions in standard atmosphere (temperature: 25 ± 2 °C, and relative humidity: 60 ± 5%). The required containers were withdrawn after 0, 3, and 6 months in triplicate for analysis. The four ounce amber glass containers were also subjected to accelerated storage conditions at 40 ± 2 °C and with a relative humidity of 75 ± 5%, and the samples were withdrawn after 0, 3 and 6 months of storage. The main active ingredients (rosmarinic acid and carvacrol) in the capsules and nano-microfiber mats were rosmarinic acid and carvacrol. Organoleptic evaluation, identification and quantified tests (by HPLC) were performed and the average weight of contents was determined to evaluate the storage influence on the stability of the capsules and mats.

### 3.11. Statistical Analysis

The results were analyzed by one-way analysis of variance (ANOVA) followed by a Tukey’s multiple comparison test with the software package Prism v. 5.04 (GraphPad Software Inc., La Jolla, CA, USA). The level of significance was taken as a value of *p* < 0.05.

## 4. Conclusions

In this study, the electrospun nano-microfiber mats with the OEE were investigated with respect to their physical characteristics: structure, compounds interaction, encapsulation efficiency, dissolution rate and release kinetic models. The content of EE100 in the electrospinning solution is a critical parameter for the structure and properties of electrospun mats. The diameter of electrospun fibers increases with the increase of EE100 polymer content in the solution. The results revealed optimal traits in OEE formulations with 24% and 32% of EE100 (E-24 and E-32 formulations).

Furthermore, encapsulation efficiency improved with the increase of EE100 polymer concentration. A higher polymer concentration resulted in electrospun mats with higher fibers diameter, smaller pores and lower free motion of active compounds, which lead to the increase of effective efficiency values.

Moreover, the dissolution rate of the more promising formulations of E-24 and E-32 showed high release of carvacrol and rosmarinic acid from both formulations. However, no significant differences between formulations and between capsule fillings were found. During storage, the formulations E-24 and E-32 showed high stability.

According to the data obtained, the dissolution rate versus time profiles of E-24 and E-32 mats and capsule fillings were shape-fitted against zero-order, first-order and Korsmeyer–Peppas kinetic models. Without exception, the first-order and Korsmeyer–Peppas models reflected the best correlation.

## Figures and Tables

**Figure 1 molecules-24-00628-f001:**
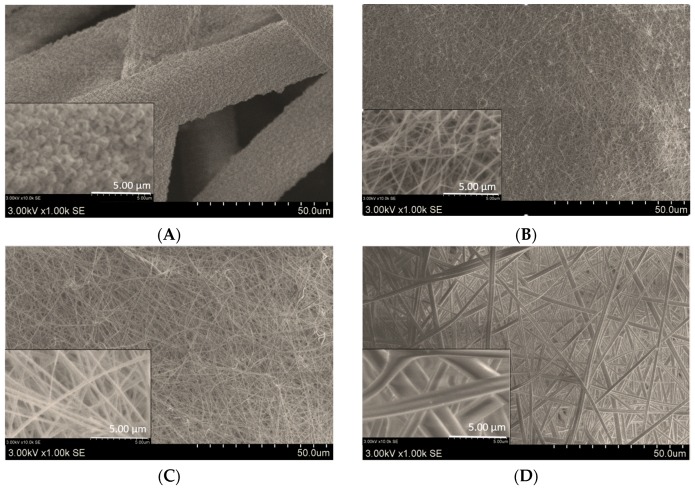
SEM images of electrospun EE100 in OEE mats at various EE100 contents (%): (**A**) E-8; (**B**) E-16; (**C**) E-24; (**D**) E-32; (**E**) E-40. Magnification: 1000×, scale bar: 50 µm, and magnification: 10,000×, scale bar: 5 µm.

**Figure 2 molecules-24-00628-f002:**
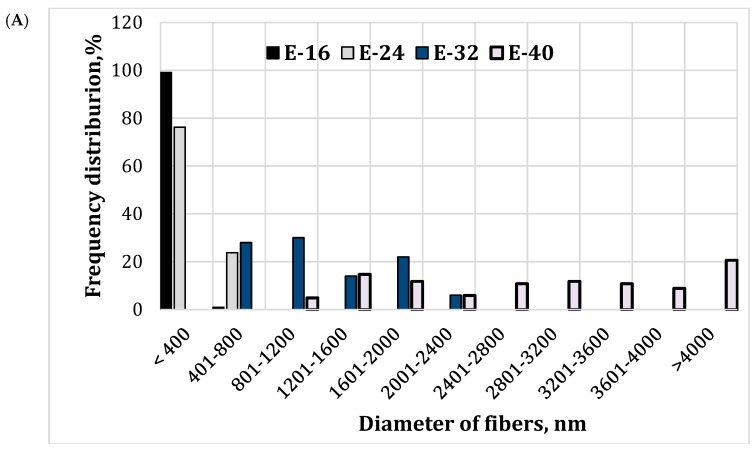
Distribution of electrospun EE100 in OEE fibers, produced from various contents of the EE100 in the OEE, as a function of the fibers diameter (nm): (**A**) E-16, E-24, E-32, and E-40, interval: 400 nm; (**B**) E-16, and E-24, interval: 150 nm.

**Figure 3 molecules-24-00628-f003:**
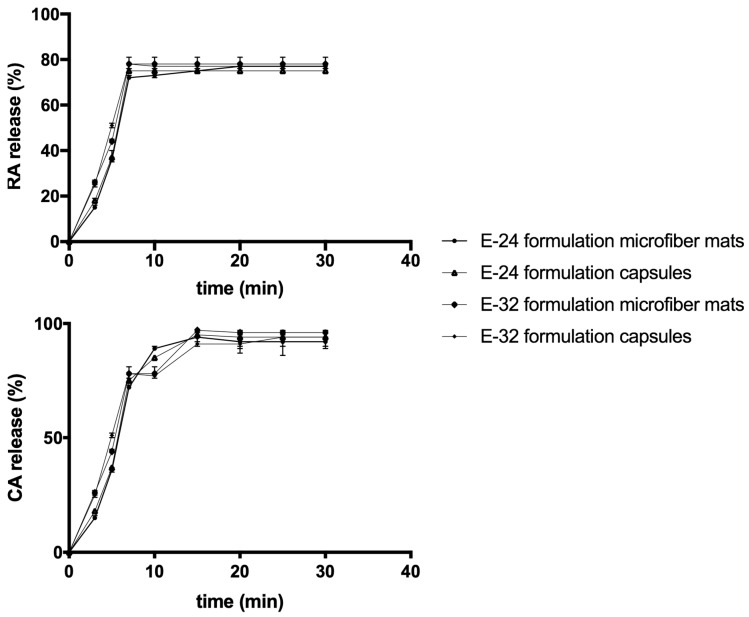
Release of rosmarinic acid and carvacrol from E-24 and E-32 nano-microfibers mats and E-24 and E-32 formulation capsules in vitro.

**Table 1 molecules-24-00628-t001:** Influence of amount of EE100 in the OEE on the solution viscosity, the conductivity and the diameter of electrospun fibers.

Sample Code	The Content of EE100 in the OEE, wt. %	Viscosity, mPa·s (±Δ)	Conductivity, µS/cm (±Δ)	Electrospun Mat	
				Average of Electrospun Fibers Diameter, nm	Standard Deviation, nm
E-8	8	2.2 ± 0.01	76.1 ± 0.87	–	–
E-16	16	34.6 ± 0.02	54.3 ± 0.33	199.7	67.6
E-24	24	79.0 ± 0.01	39.9 ± 1.89	326.1	110.0
E-32	32	263.1 ± 0.07	36.5 ± 1.65	1210.4	507.9
E-40	40	552.0 ± 0.23	36.0 ± 1.45	3101.4	1120.0

**Table 2 molecules-24-00628-t002:** Encapsulation efficiency (EE) of prepared nano-microfibers with oregano ethanolic extract.

Sample Code	EE of Carvacrol (%)	EE of Rosmarinic Acid (%)
E-8	-	-
E-16	61.91 ± 2.34	50.23 ± 1.13
E-24	87.72 ± 1.23 ^a^	70.45 ± 2.73 ^d^
E-32	91.8 ± 3.4 ^a,b^	85.6 ± 1.11 ^d,e^
E-40	67.9 ± 1.49 ^a,b,c^	60.2 ± 1.29 ^d,e,f^

^a^*p* ≤ 0.05 vs. E-16; ^b^
*p* ≤ 0.05 vs. E24; ^c^
*p* ≤ 0.05 vs. E32; ^d^
*p* ≤ 0.05 vs. E-16; ^e^
*p* ≤ 0.05 vs. E-24; ^f^
*p* ≤ 0.05 vs. E-32.

**Table 3 molecules-24-00628-t003:** Kinetic release profile model analysis of carvacrol from E-24 and E-32 formulations.

Kinetic Model	E-24 Nano-Microfiber Mats	E-24 Nano-Microfiber Capsules	E-32 Nano-Microfiber Mats	E-32 Nano-Microfiber Capsules
	**Zero-order**			
R^2^	0.830	0.896	0.787	0.847
Equation	y = 6.8494x + 12.453	y = 6.8333x + 6.8333	y = 7.0651x + 15.618	y = 7.0076x + 9.2411
	**First-order**			
R^2^	0.969	0.989	0.994	0.985
Equation	y = −0.0951x + 2.0322	y = −0.0821x + 2.0586	y = −0.138x + 2.1015	y = −0.0923x + 2.0492
	**Korsmeyer** **–** **Peppas**			
R^2^	0.990	0.986	0.983	0.979
Equation	y = 85.367x + 1.3845	y = 81.831x − 2.2099	y = 90.069x + 2.9938	y = 85.997x − 1.2791

**Table 4 molecules-24-00628-t004:** Electrospun nano-microfibers mats and capsules properties during 6 months storage under accelerated and long-term conditions (*n* = 3).

Samples	Results					
	Carvacrol Content (%)			Rosmarinic Acid Content (%)		
	Initial	3 Months	6 Months	Initial	3 Months	6 Months
A. Accelerated conditions (temperature: 40 ± 2 °C and relative humidity: 75 ± 5%).						
E-24 nano-microfiber mats	31.05 ± 0.9	30.9 ± 0.3	29.8 ± 0.3	21.89 ± 0.3	19.51± 0.8	19.46 ± 0.45
E-32 nano-microfiber mats	30.5 ± 0.01	29.7 ± 0.09	28.9 ± 0.4	20.8 ± 0.7	19.97 ± 0.2	19.09 ± 0.7
E-24 nano-microfiber capsules	30.5 ± 0.01	29.7 ± 0.09	28.9 ± 0.4	20.8 ± 0.7	19.97 ± 0.2	19.09 ± 0.7
E-32 nano-microfiber capsules	30.6 ± 0.3	29.05 ± 0.32	28.09 ± 0.17	20.6 ± 1.2	19.9 ± 0.23	19.7 ± 0.89
B. Long-term conditions (temperature: 25 ± 2 °C and relative humidity: 60 ± 5%).						
E-24 nano-microfiber mats	31.3 ± 0.2	31.04 ± 0.6	30.7 ± 0.5	19.7 ± 0.45	19.4 ± 0.78	19.0 ± 0.89
E-32 nano-microfiber mats	30.6 ± 0.3	29.6 ± 0.9	28.45 ± 0.19	20.9 ± 0.4	19.9 ± 0.11	19.45 ± 0.19
E-24 nano-microfiber capsules	29.8 ± 0.01	29.1 ± 0.76	29.0 ± 0.76	19.7 ± 0.23	19.5 ± 0.7	19.0 ± 0.54
E-32 nano-microfiber capsules	30.7 ± 0.9	29.05 ± 0.32	29.01 ± 0.31	20.2 ± 0.6	19.52 ± 0.75	19.32 ± 0.87

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
