# Peer review of "Formation and Investigation of Electrospun Eudragit E100/Oregano Mats"

_molecules, 2019, doi:10.3390/molecules24030628_

Round 1

Reviewer 1 Report

The current manuscript has described a research account on the formation and characterisation of Electrospun Eudragit E100/Oregano Mats. The authors have well-written the manuscript and discussed the obtained results, yet there are several points that need further evaluation/justification.

Physicochemical characterisation of the produced fibres could be complemented either with DSC or XRD studies to further investigate the physical state of the actives

Please justify the reason you chose  pH 1.2

The word ‘biocomponent’ should be omitted since there are no any data to support such statement

Reviewer 2 Report

Comments on "Formation and Investigation of Bicomponent Electrospun Eudragit E100/Oregano Mats"

Authors have improved the manuscript quality, changing some of the reviewers suggestions and corrections. However, manuscript still missing some main points.

As previously recommended:

1- structures were not presented in the results and discussion section;

2- revision of table and figures numbers must be carried out carefully, e.g.: page 6 - line 141, Fig 3, a must be corrected to Fig 3 (1), as presented in Figure 3. Page 8 - line 201 Figure 3 must be corrected to Fig. 4. Other table numbers must be double checked, for example page 9 - line 248.

Major points:

3- once fibers diameters were determined based on 100 measurements, statistical analysis must be carried out to determine if the results are indeed significantly different or not. Thus, the question whether claims have merit remain questionable.

4- SEM images must present a clear scale bar.

5- FTIR results must be removed and authors must double check the terms and differences between band and peak. (spectra 1 and 3 are similar demonstrating that based on FTIR principles OEE cannot be detected.) Additionally, spectra numbers must be corrected place, in order to identify them.

6- statistical analysis of Table 2 data must be carried out.

7- Authors have compared the in vitro drug release from fibers and capsules. However, the reason for this comparison is not clear. Moreover, there is no difference between those release. Additionally, error bar must be added to those graphics.

8- a conclusion is presented during results and discussion section (page 11).

9- statistical analysis must be provided for Table 4 results.

10- Figure 5, same discussion for previously in vitro release profile.

Other points:

11- Table 5 must be removed and codes presented into the text.

12- Figure 6 must be removed.

13- why have authors presented reference for calibration curves? Page 15 - lines 369 and 376. Was this calibration curve previously published?

14- revise section 3.8.

15- conclusion section remove last sentence.

Round 2

Reviewer 2 Report

Manuscript can be accepted as presented.